# Epigenetic Patterns in Five-Year-Old Children Exposed to a Low Glycemic Index Dietary Intervention during Pregnancy: Results from the ROLO Kids Study

**DOI:** 10.3390/nu12123602

**Published:** 2020-11-24

**Authors:** Aisling A. Geraghty, Alex Sexton-Oates, Eileen C. O’Brien, Richard Saffery, Fionnuala M. McAuliffe

**Affiliations:** 1UCD Perinatal Research Centre, School of Medicine, University College Dublin, National Maternity Hospital, D02 YH21 Dublin 2, Ireland; aisling.geraghty@ucdconnect.ie (A.A.G.); eileenobrien103@gmail.com (E.C.O.); 2Cancer and Disease Epigenetics, Murdoch Children’s Research Institute, Melbourne, VIC, Australia; alex.sextonoates@mcri.edu.au (A.S.-O.); richard.saffery@mcri.edu.au (R.S.); 3Department of Paediatrics, University of Melbourne, Parkville, VIC 3010, Australia

**Keywords:** epigenetics, nutrition, pregnancy, fetal programming, childhood

## Abstract

A range of in utero and early-life factors can influence offspring epigenetics, particularly DNA methylation patterns. This study aimed to investigate the influence of a dietary intervention and factors in pregnancy on offspring epigenetic profile at five years of age. We also explored associations between body composition and methylation profile in a cross-sectional analysis. Sixty-three five-year-olds were selected from the ROLO Kids Study, a Randomized controlled trial Of a LOw glycemic index dietary intervention from the second trimester of pregnancy. DNA methylation was investigated in 780,501 CpG sites in DNA isolated from saliva. Principal component analysis identified no association between maternal age, weight, or body mass index (BMI) during pregnancy and offspring DNA methylation (*p* > 0.01). There was no association with the dietary intervention during pregnancy, however, gene pathway analysis identified functional clusters involved in insulin secretion and resistance that differed between the intervention and control. There were no associations with child weight or adiposity at five years of age; however, change in weight from six months was associated with variation in methylation. We identified no evidence of long-lasting influences of maternal diet or factors on DNA methylation at age five years. However, changes in child weight were associated with the methylome in childhood.

## 1. Introduction

In adults, epigenetic changes in DNA can occur as a result of exposure to various environmental factors, including diet, tobacco, physical activity, stress, drugs, and carcinogens [1,2]. It is widely understood that epigenetic modifications in utero can occur as an adaptive response to suitably prepare the fetus for the expected postnatal environment, thereby giving them the best chance at survival [3,4]. Environmental factors during fetal development have been shown to influence variable DNA methylation patterns, which have been implicated in the determination of long-term health for the offspring [5,6], although causal relationships have not been established.

Adverse conditions in utero, such as smoking exposure during pregnancy, have been shown to influence epigenetic patterns in the offspring. A comprehensive review by Knopik et al. investigating maternal smoking during pregnancy and child epigenetic patterns identified that smoking during pregnancy was associated with alterations in placental and cord blood-derived DNA methylation patterns [7]. Research has also shown that distinct markers of smoking in pregnancy persist in the methylome of offspring at five years of age [8]. This analysis replicated findings from studies carried out in both younger and older age groups, which highlight the postnatal stability of DNA methylation changes that occur in utero. According to the Developmental Origins of Health and Disease (DOHaD) concept, both under- and over-nutrition during pregnancy may program adaptations of the fetus in order to survive in an environment that may be either lacking or be abundant in particular nutrients [9]. Some of the most substantial research relating to long-lasting impacts of maternal diet during pregnancy on offspring in later life tends to focus on nutrient-depleted populations. This is clearly demonstrated by the severe energy deficiency experienced during the famine in the Dutch Hunger Winter [10]. Research on this cohort has shown differential DNA methylation patterns in offspring aged 59 years whose mothers were exposed to the famine during their pregnancy [11,12].

Recently, research has also begun to focus on the impact of over-nutrition during pregnancy and other dietary factors that are found in developed countries. In relation to the maternal metabolic environment, exposure to adverse conditions, like gestational diabetes or maternal obesity, is increasing and is estimated to affect 10–20% of pregnancies [13]. One study identified epigenome-wide changes in teenage children whose mothers had gestational diabetes during their pregnancy [14]. Altered or increased glucose levels in utero may have an impact on the developing fetal methylome, and these changes may persist as the child ages. As demonstrated previously, a low glycemic index dietary intervention was associated with changes in neonatal methylation patterns at birth [15]. A low glycemic index diet in pregnancy may provide a more stable glycemic environment for the fetus, which could result in alterations in the fetal methylome that persist during childhood. With maternal obesity in pregnancy and conditions like gestational diabetes being controlled using dietary measures, it is important to investigate the impact of dietary intake on offspring and their health later in life. Offspring exposed to maternal diabetes in utero display accelerated growth in body mass index (BMI), but only from age two onwards [16]. This demonstrates the importance of longitudinal follow-up studies, particularly with rates of obesity in young children estimated at 8–13% in developed countries like Ireland, UK, and Australia [17]. While our low glycemic index dietary intervention during pregnancy appeared to influence offspring methylation patterns at birth, research is limited as to whether these changes persist, or whether the child’s current environment becomes a more influential factor as they age.

Candidate gene studies have identified areas of the methylome that appear sensitive to modulation and were associated with childhood body composition. Research has shown that methylation patterns in candidate genes measured in cord blood were associated with body size in children at nine years of age [18,19]. These studies, however, did not investigate whether these epigenetic marks were maintained in the methylome of the children at this age, and genome-wide research is lacking, particularly in early childhood. Research carried out in Finland identified differences in salivary DNA methylation associated with body mass index (BMI) in 11-year-old females [20]. They identified over 100 specific CpG sites (regions of DNA where a cytosine nucleotide is followed by a guanine nucleotide), with differential methylation between participants on the 10th and 90th centile for BMI. Genome-wide studies carried out in adults from multiple cohorts identified associations between DNA methylation and increased BMI [21]. In order to gain a better understanding of the pathways involved, linking epigenetic patterns with child body composition and adiposity, similar research is required in young children.

We hypothesised that the maternal environment in pregnancy, including a low glycemic index dietary intervention, would have a lasting impact on DNA methylation patterns in five-year-old children. We also hypothesised that child growth velocity and body composition is associated with variation in DNA methylation at five years of age.

## 2. Materials and Methods 

### 2.1. Study Population 

Sixty-three five-year-old children were selected from the ROLO Kids Study, a longitudinal follow-up of the ROLO Study (randomized control trial of low glycemic index diet versus no dietary intervention to prevent recurrence of fetal macrosomia [22]). In the original study, secundigravida pregnant women aged 18 years or above who had previously given birth to a macrosomic infant (birth weight >4 kg) were recruited before the 18th week of gestation. Exclusion criteria included any underlying medical disorders requiring medication (including gestational diabetes) or those not having a singleton pregnancy. After obtaining written consent, participants were randomized into either the control or intervention arm of the study. The control group received routine antenatal care and no specific dietary advice, while the intervention arm received dietary advice regarding healthy eating and, specifically, about a low glycemic index diet, which they were advised to follow for the duration of the pregnancy. The primary outcome was a change in birth weight, and no differences were noted between the control and intervention groups, though neonatal adiposity was slightly less in the intervention group [22,23]. However, significant maternal benefits were noted in terms of less gestational weight (12.2 kg vs. 13.7 kg, *p* < 0.05) and improved glucose homeostasis (glucose intolerance measured by glucose challenge testing was 25% vs. 28%, *p* < 0.05). 

Maternal height and weight were measured at approximately 14 weeks’ gestation by a trained healthcare professional, and BMI (kg/m^2^) was calculated. Detailed health and lifestyle questionnaires were collected. Three-day food diaries were completed in each trimester of pregnancy, one before the intervention and two after. The diaries were entered by the research dietitian into nutritional analysis software NetWISP version 3.0 (Tinuviel software, Llanfechell, Anglesey, UK). This software utilizes the food composition database from the sixth edition of McCance and Widdowson’s food composition tables [24]. Dietary glycemic index in each trimester was calculated using NetWISP version 3.0 based on values using the 2008 International Tables of Glycemic Index Values and more recently published values as updated in 2011 [25]. Mothers were invited to participate in the follow-up ROLO Kids Study when their child turned six months, two years, and five years of age (numbers attending at each time point are detailed in Table 1). 

### 2.2. Dietary Intervention During Pregnancy

The intervention consisted of one group dietary education session with a research dietitian, lasting approximately two hours and given in trimester two of pregnancy. The advice involved choosing as many low glycemic index foods as possible and replacing high glycemic index foods with low glycemic index alternatives. The recommended diet was eucaloric, and participants were not advised to reduce their current calorie intakes. After the education session, the participants received written resources about low glycemic index foods, and they met with the dietitian again at 28 and 34 weeks’ gestation to reinforce the dietary advice. 

### 2.3. Anthropometry and Body Composition

At birth, the children were weighed by a midwife and gestational age was noted. The children attended follow-up appointments with their mother at six months, two years, and five years of age. Weight and height were measured at each time point and BMI was calculated (kg/m^2^). Abdominal circumference (cm) was measured using a SECA® measuring tape, and skinfold thicknesses at the bicep, tricep, subscapular, and thigh was measured using calipers (mm). From these measurements, the waist to height ratio, sum of skinfolds, and subscapular to tricep ratio were calculated. Weight and BMI values were converted to standardized scores (BMI-SDS) and centiles relative to 1990 UK reference data using the Excel LMS Growth macro [26,27] for each age. A saliva sample was collected at the five year appointment using an Oragene self-collection DNA kit (OG-500 DNA Self-Collection Kit, Genotek, Ottawa, ON, Canada) under the supervision of a trained researcher. 

### 2.4. DNA Extraction and Genome-Wide Methylation Detection

Saliva samples were collected and stored at room temperature until they were extracted using the Prepito DNA Cyto Pure Kit, CMG-2034 on the Chemagic Prepito-D^®^ (PerkinElmer Inc., Wallac Oy, Turku, Finland). The purified DNA was then bisulfite converted (EZ-96 DNA Methylation-LightningTM MagPrep kit, Irvine, CA, USA), and DNA methylation was measured using the Illumina MethylationEPIC Array (EPIC, Illumina, San Diego, CA, USA), carried out by ServiceXS in Leiden, the Netherlands. Data was processed using the lumi and minfi packages for R and normalized using SWAN [28]. Probes located on sex chromosomes, those that failed in one or more samples, and those with SNPs with a minor allele frequency >1% were excluded. This left 780,501 probes for analysis in the 63 samples. 

### 2.5. Statistical Analysis

Participant characteristics were assessed for normality by visual analysis of histograms. Relationships between the central tendencies were examined using paired sample t-tests or Mann–Whitney U. Significance was set at a *p* < 0.05. Principal component and linear regression analysis was performed using the WGCNA and limma packages for R [29]. Principal component analyses were used to identify associations of variation in DNA methylation with maternal and child factors and exposure to the dietary intervention using a significance of *p* < 0.01. Linear regression analysis comparing DNA methylation between control and intervention children incorporated chip position and child sex as covariates. The Benjamini–Hochberg false discovery rate method [30] was used to adjust for multiple testing when defining statistically differentially methylated probes (adjusted *p* < 0.05). 

### 2.6. Pathway Analysis

Gene ontology analysis was carried out on annotated genes associated with the top 1000 differentially methylated CpG sites between intervention and control groups using DAVID (Database for Annotation, Visualization and Integrated Discovery v6.87) [31,32], and functional clusters were created using KEGG and Reactome Pathways.

## 3. Results

### 3.1. Cohort Characteristics and the Dietary Intervention

There were no significant differences in the maternal characteristics, such as ethnicity, education, age, and weight during pregnancy, between the intervention and control groups (Table 1). No difference was seen in the mean daily glycemic index in trimester one (prior to the intervention), however, the women in the intervention group significantly reduced their daily glycemic index in trimester two and three of pregnancy (*p* = 0.022 and 0.013, respectively). There were no significant differences in weight, height, or body composition at five years of age in children born to mothers in the intervention or control group (Table 2). At five years of age, 22.22% of children were in the overweight category (*n* = 14, BMI centile ≥85), and 6.35% were in the obese category (*n* = 4, BMI centile ≥95). One child (1.59%) had a BMI centile >5, in the underweight category. This did not differ between the control or intervention groups. 

### 3.2. Principal Component Analysis

Principal Component Analysis (PCA) was conducted to examine sources of variation within the EPIC methylation dataset. Figure 1 visualizes the top 10 principal components (PCs) contributing to variation in our cohort. This revealed that position on the array chip, child sex, and the individual were contributing significantly to the variation within the data (Table 3). Maternal factors and environment in pregnancy were not associated with variation in methylation in the five-year-old children. Birth weight or gestational age were not associated (*p* > 0.05).

### 3.3. Low Glycemic Index Dietary Intervention

PCA analysis was carried out on a reduced number of 60 participants for which glycemic index was available in each trimester. The PCA identified no association between control or intervention groups with offspring methylation status at five years of age (Table 3). Daily glycemic index values in each trimester were not strongly associated with the top 10 PCs created for variation in the methylation status of the five-year-old offspring (Table 4). Trimester 1 glycemic index was associated with PC9, representing a minimal amount of variation (r = −0.264, *p* = 0.041). Using linear regression analysis, no significantly differentially methylated probes were identified between the two groups using an adjusted *p* < 0.05. However, there were 22,181 differentially methylated probes with an unadjusted *p* < 0.01. The top 1000 highest ranked CpG sites different between the intervention and control groups, controlling for child sex and array position, were selected for gene pathway analysis. For all probes, the average beta value in the control samples was 0.629. For the intervention samples, the average was 0.628. In the top 1000 probes, the average beta value for the control samples was 0.541 and for the intervention samples was 0.551. A total of 665 of the top 1000 probes had increased methylation in the intervention samples compared to the control samples, with an average increase of 4% in methylation. Of the top 1000 differentially methylated probes, 335 showed decreased average methylation in the intervention group relative to control samples, with an average decrease of 4% in methylation per CpG site; 665 probes gained methylation in the intervention group, while 335 probes lost methylation in the intervention group (see Figure 2).

### 3.4. Pathway Analysis 

Functional clusters based on the top 1000 differentially methylated probes between the control and intervention groups were created using DAVID. We selected the top three clusters created based on KEGG and REACTOME pathways, which are summarized in Table 5. All three clusters reached a significant enrichment score of ≥1.3. The top cluster was related to insulin functioning. The genes from this cluster are involved in pathways relating to insulin resistance, insulin signaling, AMPK (adenosine monophosphate-activated protein kinase) signaling, and FOXO signaling.

### 3.5. Early Childhood Weight Status and Adiposity

Principal component analysis was carried out examining associations between child body composition at five years (Appendix A) and weight status at six months and two years (Table 6) with the DNA methylation profile. This revealed no strong associations between child weight, height, BMI, or adiposity and methylation status at five years of age (*p* > 0.5). Change in weight centiles from birth to five years was associated with PC1 and, in a reduced sample set, weight at six months of age was associated with PC6 (Table 6). No other measures of weight or adiposity were associated with methylation status at five years of age.

## 4. Discussion

We found no association of maternal factors during pregnancy with offspring DNA methylation status at five years at age. Child weight at birth and gestational age were not associated with methylation status, and limited evidence for a lasting impact of the intervention was identified at this age. Body composition at five years was not associated with methylation status, however, change in weight centile from birth to five years of age was associated with variation in the sample set, which suggests that growth patterns may be linked with the methylome in childhood. 

The lack of associations between maternal factors, particularly maternal weight or BMI, and child methylation patterns at five years of age suggests that these factors may not have a lasting impact on the methylome. Previous research has found some differences in methylation patterns in offspring at birth based on maternal BMI during pregnancy [33], and these changes may persist as the child ages, impacting later body composition. However, more recent research carried out an erudite investigation into maternal adiposity during pregnancy and concluded that maternal BMI in pregnancy and subsequent offspring adiposity is explained by genetic transmission of BMI-associated gene variants [34], as opposed to pervasive changes in DNA methylation. Longitudinal analysis of DNA methylation associated with birth weight and gestational age was also carried out from birth to adolescence, in which a lack of persistence of DNA methylation differences beyond early childhood was observed [35].

Despite the low glycemic index diet being associated with variation in DNA methylation in newborns in this cohort [15], there was no association of the intervention group with DNA methylation profiles at five years of age. Interestingly, however, maternal glycemic index in trimester 1 of pregnancy was significantly associated with some variation in the child DNA methylation profile. Studies have previously shown a lasting impact of a low glycemic index diet in offspring, with associations identified between dietary glycemic index in pregnancy and markers of metabolic syndrome in the offspring at 20 years of age [36]. It may be that more persistent changes in DNA methylation are triggered when the fetus is exposed to the stimuli in the earliest stage of development at the beginning of pregnancy. The Dutch Hunger Winter study has shown that the timing of the exposure is critical, with evidence suggesting that early gestation (before 10 weeks) is uniquely sensitive to pervasive changes in offspring DNA methylation [12]. So while dietary glycemic index in trimester 1 was associated with a small amount of variation in DNA methylation patterns of the offspring at five years of age, our low glycemic index intervention took place at approximately the thirteenth week of gestation, which may have been too late in fetal development to have a long-lasting impact on the methylome in later life. The Dutch Hunger Winter study concluded that early gestation, but not mid or late gestation, was critical for detectable changes in adult DNA methylation [12]. Although associations with the dietary intervention weren’t statistically significant, exploratory pathway analysis into the top 1000 differentially methylated CpG sites between the control and intervention groups revealed some intriguing findings. The top cluster was related to insulin functioning, a pathway plausibly linked to the low glycemic index intervention. This could potentially explain the mechanism behind previous research, where a low glycemic index diet in pregnancy was reported to influence insulin functioning in the adult offspring [36]. AMPK signaling, the third pathway identified, has also been shown to have a role in metabolism pathways influencing glucose uptake [37]. Additionally, an Australian pilot randomized control trial observed that a low glycemic index diet during pregnancy influenced offspring arterial wall thickness in early childhood [38]. The second pathway we identified related to the intervention was Notch signaling, which has been shown to play a role in cardiovascular health, particularly in pediatric cardiac disease [39]. These findings may aid our understanding of the long-term impacts of a low glycemic index dietary intervention during pregnancy.

There were no associations of child weight or adiposity with methylation status at five years of age in this cohort. While previous studies have found associations with methylation and child adiposity, these tend to be gene-specific, targeted studies with methylation status at a younger age predicting body composition in later life [18,19,40]. Our results suggest that change in weight and growth patterns rather than body composition at one time point in offspring may be guided by DNA methylation, particularly in early life, with our findings of weight centiles and BMI at six months of age being associated with some variation in methylation status in a subset of the sample group. There is a very powerful link between DNA methylation and aging, suggesting a plausible mechanistic role for variable DNA methylation patterns in controlling body composition trajectories as the child ages. This is consistent with the DOHaD hypothesis, which proposes that prenatal exposures can alter developmental trajectories [41]. The DOHaD hypothesis highlights the link between the peri-conceptual period, fetal development, and early infancy with the subsequent development of metabolic disorders in later life [42]. With DNA methylation patterns at birth being associated with child adiposity in later life [18,43], research is limited as to whether these patterns persist and are measurable in later life. These patterns of methylation may resolve or change as the child ages, and further research is required to determine this. 

This study has many strengths, including the utilization of an extensive DNA methylation array, the Illumina Infinum MethylationEPIC BeadChip Array, which allowed the investigation of 780,501 CpG sites in the salivary DNA. We also had extensive comprehensive information on the participants, with each of the body measurements being collected by trained research staff, as opposed to relying on self-reported measures. We had detailed information on the participants at numerous time points, both phenotypic data along with dietary data. This allowed us to investigate the timings of associations and how this may change. This analysis was not without limitations, however. While sufficient for exploratory analysis, with the current sample size, statistical power may be limited to detect more subtle alterations in the methylome, particularly in relation to the biological impacts of the dietary intervention five years postpartum. The DNA methylation profiles were derived from saliva samples, and methylation patterns can vary based on tissue type. Other DNA sources, such as adipose tissue or blood, may be more strongly related to child body composition. However, research on adolescent females has previously shown an association between salivary DNA methylation and body size [20]. There also are various other epigenetic mechanisms, like histone modification, which may have a role here or may be more responsive to environmental stimuli. In addition, more sensitive measures of body composition, such as air displacement plethysmography or dual-energy X-ray absorptiometry, may be more suitable methods to detect subtle differences. This represents further avenues of investigation. 

These findings may serve to elucidate the factors that influence long-lasting DNA methylation patterns and the mechanisms behind which environment factors impact child body composition. With current research recognizing the vital role that epigenetics has in programming for later health [42], this research expands on this area with longitudinal findings related to children. We found that in this cohort, factors in pregnancy did not have a large or lasting impact on overall variation in DNA methylation status of offspring at five years of age. We also found that methylation patterns were not associated with current child body composition. These findings may be important in guiding future research in this area. Extensive longitudinal research is required, along with suitably powered randomized controlled trials, to investigate potentially time-sensitive long-term impacts of environmental factors during pregnancy on child DNA methylation patterns and those in later life. 

## 5. Conclusions

In this modestly-sized discovery sample, we identified limited evidence of long-lasting influences of a low glycemic index dietary intervention in pregnancy or maternal factors on child DNA methylation patterns at five years of age. While the intervention was not a primary cause of variation in the sample set, this may be due to the small sample size, and therefore limited the capacity to detect an effect. The differences in pathways between the intervention and control groups were related to insulin secretion and resistance, along with cardiac functioning. Child body composition was not related to DNA methylation patterns at this age, however, changes in weight centiles were associated, suggesting a mechanistic role in growth trajectories. Larger studies and replication of these findings are required to substantiate whether methylation differences at birth have a long-lasting effect into childhood and to investigate what factors influence the methylome in children.

## Figures and Tables

**Figure 1 nutrients-12-03602-f001:**
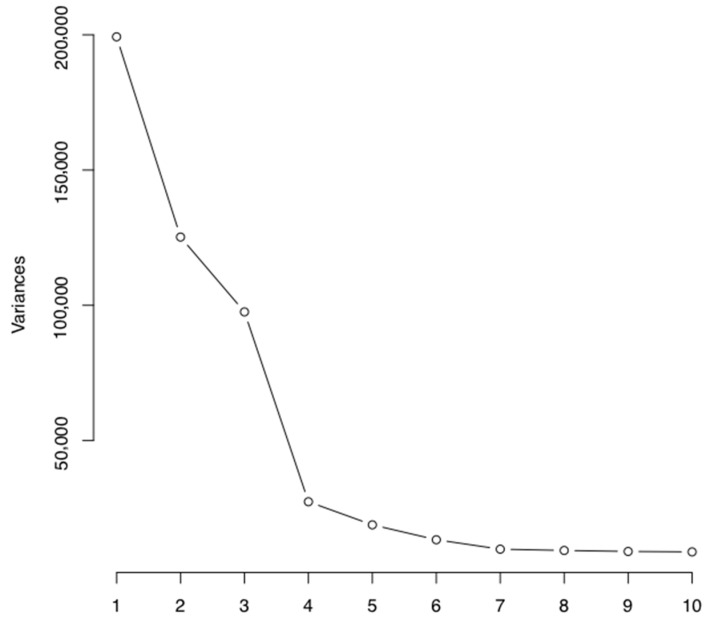
Principal component analysis plot of first 10 principal components responsible for variation in methylation in the ROLO cohort (Randomised controlled trial of a LOw glycaemic index diet in pregnancy).

**Figure 2 nutrients-12-03602-f002:**
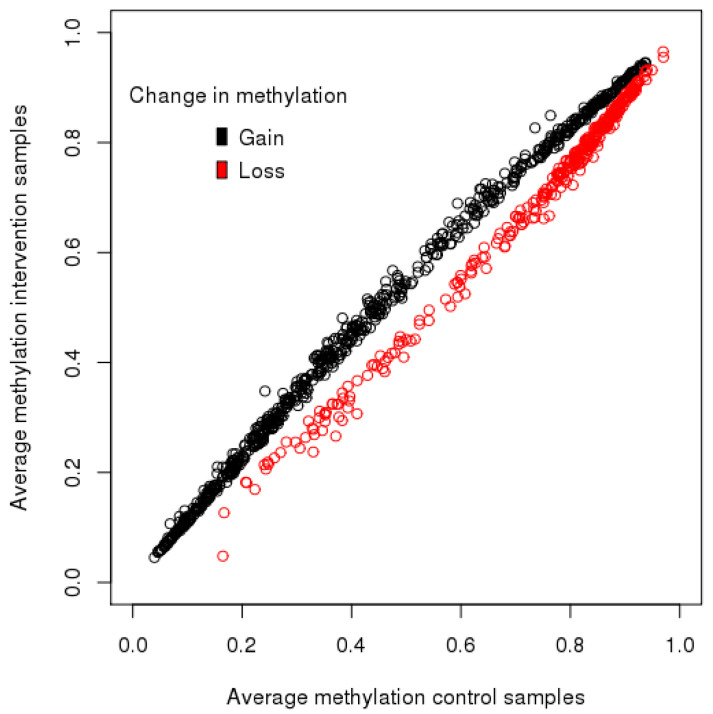
Scatter plot of the average methylation of the top 1000 differentially methylated CpG sites and the change in methylation between the intervention and control groups.

**Table 1 nutrients-12-03602-t001:** Characteristics of the mothers and offspring (up to two years of age) of the intervention and control groups in the ROLO cohort.

		Total			Intervention			Control		
Maternal Characteristics	*n*	Mean/Median	SD/IQR	*n*	Mean/Median	SD/IQR	*n*	Mean/Median	SD/IQR	*p*
Maternal age at birth (mean, SD, years)	63	33.07	3.66	31	32.88	3.99	32	31.79	7.91	0.697
Gestational age at booking visit (mean, SD, weeks)	63	13.00	2.34	31	13.23	2.45	32	12.78	2.24	0.454
Maternal weight in early pregnancy (median, IQR, kg)	63	68.5	8.8	31	69.6	10.7	32	66.2	9.38	0.375
Maternal BMI in early pregnancy (mean, SD, cm)	63	25.75	4.05	31	25.74	3.46	32	25.77	4.61	0.979
Maternal smoking in pregnancy (*n* (%))	63	1 (1.6)		31	1 (3.2)		32	0 (0)		0.987
Maternal ethnicity: Caucasian (*n* (%))	63	61 (96.8)		31	30 (96.8)		32	31 (96.9)		0.37
Education higher level (*n* (%))	59	34 (57.6)		29	18 (58.1)		30	16 (53.3)		0.678
Trimester 1 GI (mean, SD)	60	58.07	4.48	30	57.67	4.6	30	58.46	4.41	0.497
Trimester 2 GI (mean, SD)	60	56.57	3.38	30	55.58	3.05	30	57.56	3.46	0.022 *
Trimester 3 GI (mean, SD)	60	56.92	4.27	30	55.57	3.88	30	58.26	4.27	0.013 *
Child Characteristics		Mean/Median	SD/IQR		Mean/Median	SD/IQR		Mean/Median	SD/IQR	*p*
Sex: male (*n* (%))	63	32 (50.8)		31	16 (51.6)		32	16 (50)		1
Birth weight (mean, SD, kg)	63	4.06	0.47	31	4.14	0.52	32	3.98	0.41	0.173
Birth weight centile (median, IQR)	63	79.55	33.43	31	82.75	34.18	32	76.15	34.23	0.379
Gestational age (median, IQR, weeks)	63	40.57	1.57	31	40.43	1.57	32	40.714	2.18	0.94
Breastfed (*n* (%))	39	27 (69.2)		20	16 (80)		19	11 (57.9)		0.251
Six Month Follow-Up										
Weight (mean, SD, kg)	50	8.57	1.27	23	9.03	1.49	27	8.19	0.91	0.024 *
Weight centile (median, IQR)	50	72.5	39.25	23	84	46	27	67	54	0.078
BMI centile (median, IQR)	50	51	57.25	23	74	62	27	40	50	0.016 *
Two Year Follow-Up										
Weight (mean, SD, kg)	56	13.15	1.37	29	13.4	1.48	27	12.92	1.22	0.216
Weight centile (median, IQR)	56	67	44	29	77	48.5	27	63	41	0.298
BMI centile (median, IQR)	56	41.5	48.5	29	56	66	27	38	32	0.275

Normally distributed data are given as mean and SD, and non-normally distributed data are given as median and IQR. BMI: body mass index; GI: glycemic index; SD: standard deviation; SDS: Standard deviation scores; IQR: interquartile range. Statistical comparisons done by Student’s t-test, Mann–Whitney U, or chi-square tests. * Significant at *p* < 0.05.

**Table 2 nutrients-12-03602-t002:** Characteristics of offspring born into the intervention and control groups in the ROLO cohort at five years of age.

		Total			Intervention			Control		
Child Characteristics at the Five Year Follow-Up	*n*	Mean/Median	SD/IQR	*n*	Mean/Median	SD/IQR	*n*	Mean/Median	SD/IQR	*p*
Age (mean, SD, years)	63	5.10	0.11	31	5.08	0.12	32	5.12	0.1	0.134
Weight (mean, SD, kg)	63	20.28	2.33	31	20.43	2.49	32	20.13	2.21	0.624
Weight centile (median, IQR)	63	71	45	31	70	48	32	74.5	20.5	0.625
Change in weight centiles from birth to five years (mean, SD)	63	8.9	30.95	31	9.94	30.53	32	7.9	31.79	0.799
Height (mean, SD, cm)	63	111.26	4.13	31	110.98	4.12	32	111.53	4.19	0.599
Height centile (median, IQR)	63	63	47	31	63	48	32	65	47.75	0.67
BMI (mean, SD, kg/m^2^)	63	16.35	1.32	31	16.55	1.46	32	16.16	1.15	0.237
BMI centile (median, IQR)	63	69	39	31	71	35	32	68.5	40.75	0.437
Chest circumference (mean, SD, cm)	63	56.44	2.62	31	57.05	2.61	32	55.84	2.53	0.066
Abdomen circumference (mean, SD, cm)	63	54.41	5.61	31	55.76	4.15	32	53.11	6.53	0.059
Waist to height ratio (mean, SD)	63	0.49	0.05	31	0.5	0.03	32	0.48	0.07	0.068
Sum of skinfolds (median, IQR, mm)	55	34.8	13.6	28	35.25	14.48	27	33.5	10	0.853
Subscapular and Triceps skinfold (median, IQR, mm)	55	15.28	3.89	27	15.89	4.6	28	14.69	3.15	0.256
Subscular:triceps skinfold ratio (mean, SD)	55	0.62	0.16	27	0.64	0.21	28	0.61	0.13	0.16

Normally distributed data are given as the mean and SD, and non-normally distributed data are given as the median and IQR. BMI: body mass index; GI: glycemic index; SD: standard deviation; IQR: interquartile range. Statistical comparisons were done with Student’s *t*-test, Mann–Whitney U, or chi-square tests.

**Table 3 nutrients-12-03602-t003:** Principal component analysis of maternal and child factors during pregnancy and DNA methylation levels in five-year-old offspring (*n* = 63).

	Individual	Chip	Array Position	RCT Group	Maternal Age (years)	Maternal Weight (kg)	Maternal BMI	Smoking in Pregnancy	Maternal Ethnicity	Child Sex	Gestational Age (weeks)	Birth Weight (kg)
PC1 correlation	−0.056	0.131	0.392 *	0.163	−0.094	0.064	0.016	−0.175	0.087	−0.076	−0.056	−0.084
PC1 *p*-value	0.662	0.306	0.002 *	0.202	0.465	0.620	0.904	0.171	0.497	0.556	0.665	0.512
PC2 correlation	−0.047	−0.002	0.232	−0.194	−0.069	−0.103	−0.084	0.015	−0.011	0.040	0.241	−0.116
PC2 *p*-value	0.713	0.985	0.067	0.128	0.590	0.421	0.515	0.905	0.929	0.757	0.058	0.364
PC3 correlation	−0.110	−0.137	0.685 *	−0.046	−0.035	0.088	0.078	0.055	0.073	0.073	−0.167	−0.018
PC3 *p*-value	0.392	0.283	0.000 *	0.720	0.783	0.495	0.543	0.671	0.571	0.568	0.190	0.887
PC4 correlation	−0.101	0.234	0.083	−0.154	0.003	0.064	0.111	−0.053	0.069	−0.002	0.041	0.152
PC4 *p*-value	0.430	0.065	0.519	0.228	0.982	0.620	0.385	0.681	0.589	0.989	0.750	0.234
PC5 correlation	−0.296 *	0.316 *	0.151	0.012	−0.016	0.018	0.007	−0.099	0.010	−0.042	−0.148	−0.035
PC5 *p*-value	0.019 *	0.012 *	0.237	0.923	0.903	0.890	0.957	0.438	0.935	0.746	0.247	0.786
PC6 correlation	−0.132	0.109	−0.147	−0.192	−0.191	−0.044	−0.011	−0.013	−0.068	−0.303 *	−0.018	0.155
PC6 *p*-value	0.304	0.394	0.251	0.132	0.133	0.730	0.929	0.921	0.594	0.016 *	0.886	0.225
PC7 correlation	0.048	0.179	0.051	−0.078	−0.039	−0.125	−0.152	0.011	−0.029	−0.362 *	−0.030	0.127
PC7 *p*-value	0.709	0.160	0.689	0.545	0.762	0.328	0.235	0.929	0.822	0.004 *	0.816	0.323
PC8 correlation	−0.117	−0.210	0.065	0.009	0.039	0.018	0.047	−0.013	0.070	0.198	0.246	−0.088
PC8 *p*-value	0.360	0.099	0.612	0.944	0.761	0.890	0.713	0.918	0.583	0.120	0.052	0.494
PC9 correlation	0.035	0.155	0.000	−0.057	0.146	−0.176	−0.167	−0.011	−0.009	0.009	−0.058	−0.064
PC9 *p*-value	0.787	0.225	1.000	0.658	0.255	0.169	0.192	0.931	0.944	0.944	0.652	0.621
PC10 correlation	0.055	0.142	0.067	0.236	0.013	−0.168	−0.155	−0.012	0.023	−0.223	0.035	0.139
PC10 *p*-value	0.668	0.268	0.602	0.063	0.921	0.188	0.225	0.924	0.856	0.080	0.787	0.278

BMI: body mass index; PC: principal component; RCT: randomized control trial. * Significant at *p* < 0.05.

**Table 4 nutrients-12-03602-t004:** Principle component analysis of dietary glycemic index during pregnancy and DNA methylation levels in five-year-old offspring (*n* = 60).

	Individual	T1 GI	T2 GI	T3 GI
PC1 correlation	−0.074	−0.008	−0.013	0.024
PC1 *p*-value	0.574	0.951	0.923	0.857
PC2 correlation	0.057	0.034	0.167	0.178
PC2 *p*-value	0.667	0.799	0.203	0.173
PC3 correlation	0.139	−0.033	−0.033	0.185
PC3 *p*-value	0.288	0.802	0.802	0.157
PC4 correlation	−0.077	0.042	−0.050	−0.072
PC4 *p*-value	0.559	0.749	0.702	0.584
PC5 correlation	−0.336 *	0.158	0.208	−0.016
PC5 *p*-value	0.009 *	0.229	0.112	0.906
PC6 correlation	−0.108	0.191	−0.037	0.034
PC6 *p*-value	0.411	0.144	0.781	0.796
PC7 correlation	−0.021	0.135	0.114	0.211
PC7 *p*-value	0.872	0.303	0.384	0.105
PC8 correlation	0.135	−0.091	−0.010	−0.156
PC8 *p*-value	0.302	0.488	0.940	0.235
PC9 correlation	−0.033	−0.264 *	−0.105	0.030
PC9 *p*-value	0.803	0.041 *	0.424	0.820
PC10 correlation	−0.051	−0.159	−0.081	−0.218
PC10 *p*-value	0.700	0.225	0.538	0.094

GI: glycemic index; PC: principal component; T1: trimester 1; T2: trimester 2; T3: trimester 3. * Significant at *p* < 0.05.

**Table 5 nutrients-12-03602-t005:** Gene functional clusters of the 1000 highest ranked probes from linear regression analysis associated with membership of the ROLO intervention or control group.

	Pathway Type	Function	Count	*p*	Benjamini
**Cluster 1: Insulin Functioning**					
**ES: 1.91 ***	KEGG	Insulin resistance	11	0.0023	0.17
	KEGG	AMPK signaling pathway	11	0.0055	0.21
	KEGG	Insulin signaling pathway	11	0.013	0.25
	KEGG	FoxO signaling pathway	8	0.14	0.56
**Cluster 2: NOTCH Functioning**					
**ES: 1.83 ***	REACTOME	Constitutive signaling by NOTCH1 HD+PEST domain mutants	8	0.0017	0.25
	REACTOME	Constitutive signaling by NOTCH1 PEST domain mutants	8	0.0017	0.25
	REACTOME	NOTCH1 intracellular domain regulates transcription	6	0.014	0.65
	KEGG	Notch signaling pathway	6	0.018	0.29
	REACTOME	Notch-HLH transcription pathway	3	0.059	0.85
	REACTOME	Pre-NOTCH transcription and translation	3	0.22	0.98
**Cluster 3: Cell Signal Functioning**					
**ES: 1.31 ***	REACTOME	PIP3 activates AKT signaling	8	0.012	0.71
	REACTOME	CD28 dependent PI3K/Akt signaling	4	0.029	0.77
	REACTOME	Constitutive signaling by AKT1 E17K in cancer	4	0.04	0.8
	REACTOME	VEGFR2 mediated vascular permeability	3	0.22	0.98

Clusters created using the DAVID (http://david.abcc.ncifcrf.gov/) Gene Functional Classification Tool and KEGG and REACTOME pathways. ES: enrichment score; Benjamini: Benjamini–Hochberg false discovery rate method; AKT: protein kinase B; AMPK: adenosine monophosphate-activated protein kinase; CD28: Cluster of Differentiation 28; PIP3: phosphatidylinositol-3,4,5-triphosphate; PI3K phosphatidylinositol 3-kinase; VEGFR2: Vascular endothelial growth factor receptor 2; (p-values ≤ 0.05). * Significant ES.

**Table 6 nutrients-12-03602-t006:** Principal component analysis of early childhood weight and growth influencing DNA methylation levels at five years of age.

	Change in Weight Centiles	Weight Centile at Six Months (*n* = 50)	Weight Centile at Two Years (*n* = 56)	BMI Centile at Six Months (*n* = 50)	BMI Centile at Two Years (*n* = 56)
PC1 correlation	−0.253 *	0.119	−0.020	0.099	−0.098
PC1 *p*-value	0.047 *	0.410	0.887	0.495	0.477
PC2 correlation	−0.028	0.054	−0.056	−0.120	−0.044
PC2 *p*-value	0.826	0.711	0.683	0.405	0.751
PC3 correlation	−0.131	−0.258	0.220	−0.153	0.025
PC3 *p*-value	0.312	0.071	0.107	0.289	0.854
PC4 correlation	0.087	−0.059	−0.076	−0.035	−0.081
PC4 *p*-value	0.500	0.683	0.583	0.809	0.555
PC5 correlation	−0.082	0.040	0.021	0.056	−0.055
PC5 *p*-value	0.524	0.783	0.878	0.699	0.692
PC6 correlation	0.064	−0.283 *	−0.132	−0.386 *	−0.139
PC6 *p*-value	0.624	0.047 *	0.336	0.006 *	0.312
PC7 correlation	−0.128	−0.128	0.172	−0.174	0.188
PC7 *p*-value	0.322	0.374	0.209	0.227	0.169
PC8 correlation	0.065	−0.047	−0.017	−0.239	0.080
PC8 *p*-value	0.614	0.743	0.902	0.094	0.559
PC9 correlation	0.004	−0.339 *	0.131	−0.216	0.022
PC9 *p*-value	0.978	0.016 *	0.339	0.133	0.873
PC10 correlation	−0.030	−0.147	−0.021	−0.105	0.001
PC10 *p*-value	0.818	0.307	0.881	0.468	0.993

BMI: body mass index (kg/m^2^); PC: principal component. * Significant at *p* < 0.05.

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
