# Peer review of "Epigenetic Patterns in Five-Year-Old Children Exposed to a Low Glycemic Index Dietary Intervention during Pregnancy: Results from the ROLO Kids Study"

_nutrients, 2020, doi:10.3390/nu12123602_

Round 1

Reviewer 1 Report

This study investigates the association between maternal diet during pregnancy and epigenetic profile of the offspring at the age of 5. The researchers found no association between maternal intervention with low glycaemic diet and DNA methylation status in 5 year olds. In addition, no association was found between methylation status and body composition. It is a well written manuscript which provides new insights into the possible effects of maternal diet modifications during pregnancy on child health. I have the following comments:

  1. In the introduction section the authors indicate that “maternal obesity is estimated to affect 10-20% of pregnancies”. What is the incidence of childhood obesity in Ireland, UK and Australia?
  2. Did you obtain information regarding the dietary habits of the children? As there is “limited evidence for a lasting impact of the intervention” the dietary habits are more likely to play a role in obesity?
  3. Based on these results and on their personal clinical experience would the authors suggest low glycaemic diet to a pregnant woman who had a macrosomic infant in their previous pregnancy?
  4. Is it plausible to plan future long-term (longer than 5 years) research taking into consideration the lack of association in this study?

Author Response

Many thanks for the reviewer for their positive feedback.

1. In the introduction section the authors indicate that “maternal obesity is estimated to affect 10-20% of pregnancies”. What is the incidence of childhood obesity in Ireland, UK and Australia?

Response to reviewer: Thank you for this comment. Rates of obesity in young children are estimated at 8-13% in Ireland, UK, and Australia. We have added this information and the following reference into the introduction: [17] NCD Risk Factor Collaboration (NCD-RisC). Worldwide trends in body-mass index, underweight, overweight, and obesity from 1975 to 2016: a pooled analysis of 2416 population-based measurement studies in 128·9 million children, adolescents, and adults. Lancet. 2017;390:2627–42. (See lines 67-69 and new reference 17).

2. Did you obtain information regarding the dietary habits of the children? As there is “limited evidence for a lasting impact of the intervention” the dietary habits are more likely to play a role in obesity?

Response to reviewer: Thank you for this comment. Unfortunately we did not have dietary intakes for these participants available for this analysis which would have been interesting, however, as dietary intakes are strongly influenced by socioeconomic factors we did check whether there was a difference in maternal education level (as a marker of SES) between the intervention and control groups and there was no significant difference.

3. Based on these results and on their personal clinical experience would the authors suggest low glycaemic diet to a pregnant woman who had a macrosomic infant in their previous pregnancy?

Response to reviewer: Thank you for this comment. Our findings show that a low GI diet in pregnancy is safe and has no negative impacts on the mother or the child, in fact there are significant benefits to the mother during pregnancy in terms of less gestational weight (12.2kg vs 13.7kg, P< 0.05) and improved glucose homeostasis (glucose intolerance 25% vs 28% P<0.05). We highlight this in the methods when discussing the study (see lines 104-106).

4. Is it plausible to plan future long-term (longer than 5 years) research taking into consideration the lack of association in this study?

Response to reviewer: Thank you for this comment. Yes, we believe that it is still important to carry out long-term research as other studies have found that some differences are not detectable until later in life, such as Crume et al, 2011 who found that offspring exposed to diabetes in-utero did not display differences in BMI trajectories until age 2 onwards. Longitudinal research is vital to identify potentially time-sensitive impacts of environmental factors during pregnancy. We have included this point in the discussion (see discussion, lines 338-340).

Reviewer 2 Report

Geraghty and colleagues evaluate the effects of a low glycemic index dietary intervention during pregnancy on offspring DNA methylation profile at 5 years of age. Overall, the research is interesting and enriches the current literature on fetal programming, but the manuscript needs some revision.

From a clinical point of view, why the evaluation of changes in the DNA methylation profile in children 5 years old?

Why was there no comment in the discussion of changes in the AMPK signaling pathway?

The title of the manuscript should contain information on what type of dietary intervention during pregnancy was assessed.

It should be emphasized in the abstract that the dietary intervention concerned the 2nd and 3rd trimester of pregnancy.

This statement in the abstract is confusing because the p-value shows a significant effect. Please check it carefully. Principal components analysis identified no association between maternal age, weight, or BMI during pregnancy and offspring DNA methylation (p<0.01).

What factors do the authors mean (We identified no evidence of long-lasting influences of maternal diet or factors on DNA methylation at age 5 years)? It will be easier for readers if they are mentioned.

Experiments provide a rich dataset that could be of interest to a large number of researchers. Has the entire dataset from the array been deposited in an online repository for public access?

Lines 14 and 65: in-utero please italicize as in the rest of the manuscript.

Line 66: Please explain the abbreviation BMI when first used in the text (next in line 77 use only the abbreviation BMI).

Line 78: Please explain CpG abbreviation.

Line 83: Please explain what dietary intervention was assessed in the study.

Table 1. What are the reasons for differences in n different time points (6 months, 2 and 5 years of age)?

Table 2. In the description of Table 2, I suggest removing * Significant at P <0.05, because there are no statistically significant changes in the presented results.

Line 220: Please explain AMPK abbreviation.

References no 3, 17, 30, 31: Please correct authors' initials.

References no 29: Editorial correction required.

References no 41: Please correct the information about the journal.

Author Response

Many thanks for your positive feedback and helpful comments.

- From a clinical point of view, why the evaluation of changes in the DNA methylation profile in children 5 years old?

Response to reviewer: Thank you for this comment. Subtle differences were identified in DNA methylation in the offspring at birth and as there were no identifiable differences in body composition at age 5 we wanted to evaluate whether these differences in DNA methylation patterns were still evident at 5 years of age. DNA methylation patterns may inform us about differences, such as with insulin functioning, in advance of it becoming a problem so that interventions and care pathways could be put in place.

- Why was there no comment in the discussion of changes in the AMPK signaling pathway?

Response to reviewer: Thank you for this comment. We have added into the discussion that AMPK signalling has been shown to have a role in metabolism pathways influencing glucose uptake, and added in the additional reference: [37] Mihaylova M, Shaw R. The AMPK signalling pathway coordinates cell growth, autophagy and metabolism. Nat Cell Biol. 2011;13: 1016–1023. doi:10.1038/ncb2329

(see lines 289-291 and new reference 37).

- The title of the manuscript should contain information on what type of dietary intervention during pregnancy was assessed.

Response to reviewer: Many thanks for your comment. We have changed the title of the manuscript to Epigenetic patterns in 5 year old children exposed to a low Glycaemic Index dietary intervention during pregnancy: results from the ROLO Kids study. (See title, lines 2-3).

- It should be emphasized in the abstract that the dietary intervention concerned the 2nd and 3rd trimester of pregnancy.

Response to reviewer: Thank you for this comment. We have clarified in the abstract that the low glycemic index dietary intervention was from trimester 2 of pregnancy onwards. Sixty-three 5-year-olds were selected from the ROLO Kids study, a randomised controlled trial of a low glycemic index dietary intervention from the second trimester of pregnancy. (See abstract, lines 18-19)

- This statement in the abstract is confusing because the p-value shows a significant effect. Please check it carefully. Principal components analysis identified no association between maternal age, weight, or BMI during pregnancy and offspring DNA methylation (p<0.01).

Response to reviewer: Thank you for this comment. Apologies for this error, it was meant to read p>0.01. We have fixed this error in the manuscript (see abstract, line 22).

- What factors do the authors mean (We identified no evidence of long-lasting influences of maternal diet or factors on DNA methylation at age 5 years)? It will be easier for readers if they are mentioned.

Response to reviewer: Thank you for this comment. In the abstract we specify that no associations were identified between maternal age, weight, or BMI during pregnancy and offspring DNA methylation (see abstract, line 20-22).  

- Experiments provide a rich dataset that could be of interest to a large number of researchers. Has the entire dataset from the array been deposited in an online repository for public access?

Response to reviewer: Thank you for this comment. Unfortunately due to data sharing restrictions and in line with the ethical approval guidelines for this study we were not able to share this data on an online repository for public access. However, interested researchers are welcome to contact the corresponding author who may be able to make the data available upon reasonable request.

- Lines 14 and 65: in-utero please italicize as in the rest of the manuscript.

Response to reviewer: Thank you for this comment, we have made this change in the manuscript (see abstract, line 14 and introduction line 65).

- Line 66: Please explain the abbreviation BMI when first used in the text (next in line 77 use only the abbreviation BMI).

Response to reviewer: Thank you for this comment, we have explained BMI in the manuscript (see introduction, line 67).

- Line 78: Please explain CpG abbreviation.

Response to reviewer: Thank you for this comment, we have clarified that a CpG site is a region of DNA where a cytosine nucleotide is followed by a guanine nucleotide (see introduction, line 79-81).

- Line 83: Please explain what dietary intervention was assessed in the study.

Response to reviewer: Thank you for this comment, we have clarified that the dietary intervention was a low glycaemic dietary intervention (see introduction, line 86-88).

- Table 1. What are the reasons for differences in n different time points (6 months, 2 and 5 years of age)?

Response to reviewer: Thank you for this comment. Not all participants in this study were followed up at the 6 month and 2 year timepoint as DNA samples were collected at the 5 year follow-up. This has been clarified in the methods. (See methods, line 115-117 ; Mothers were invited to participate in the follow-up ROLO Kids study when their child turned 6 months, 2 years, and 5 years of age (numbers attending at each timepoint are detailed in table 1).

- Table 2. In the description of Table 2, I suggest removing * Significant at P <0.05, because there are no statistically significant changes in the presented results.

Response to reviewer: Thank you for this comment, we have removed this from Table 2 (see table 2, line 180).

- Line 220: Please explain AMPK abbreviation.

Response to reviewer: Thank you for this comment. We have explained the AMPK abbreviation in the manuscript; AMPK:adenosine monophosphate-activated protein kinase. (see lines 236-237).

- References no 3, 17, 30, 31: Please correct authors' initials.

Response to reviewer: Thank you for this comment and apologies for these typos. The initials have been fixed in the reference list (see references 3,18,31, 32).

- References no 29: Editorial correction required.

Response to reviewer: Thank you for this comment. We have fixed this reference which is now reference 30 due to the addition of a reference to the introduction (see reference list, line 443-444).

- References no 41: Please correct the information about the journal.

Response to reviewer: Thank you for this comment. We have corrected the journal information for this reference (now reference 43, line 477-479).

Reviewer 3 Report

The present work provides a new piece of evidence in favour of the absence of long-lasting epigenetic effects of lower a glycaemic intake during trimesters 2 and 3 of pregnancy. The novelty and originality of this work reside in the statistical analysis integration of all the data collected from children and mothers participating in this study, together with the genome-wide methylation studies performed.

The global message of the manuscript is clear and brings solid pieces of evidence and data. Additionally, the authors are aware of the limitations of the study.  However, it is intriguing whether the results would have been the same if the dietary intervention would have started in the first months of pregnancy.

Minor comments

  1. Figure 1 is missing. Figure 1, referred to in the paragraph 3.2 is missing. Top 10 principal components cannot be seen.
  2. The rest of the figures need to correct their numeration.

Author Response

Many thanks to the reviewer for their supportive feedback about this work. We hope it encourages further research in this area.

Minor comments

  1. Figure 1 is missing. Figure 1, referred to in the paragraph 3.2 is missing. Top 10 principal components cannot be seen.
  2. The rest of the figures need to correct their numeration

Response to reviewer: Thank you for bringing this to our attention and our apologies for this error. We have included figure 1 and corrected the other figure to Figure 2 (see corrected figures 1, lines 189-201, and figure 2, lines 222-223, in manuscript).

Reviewer 4 Report

I congratulate with the authors for excellent and hard work done (e.g. data analysis) and I suggest only minor revisions to the paper.

The suggestion are the following:

Line 18, please change 'control' in 'controlled'.

Line 31-32, Please insert drugs in the environmental factors that promote epigenetic changes in DNA, for your knowledge check this article: ' Groh A, Rhein M, Buchholz V, Burkert A, Huber CG, Lang UE, Borgwardt SJ, Heberlein A, Muschler MAN, Hillemacher T, Bleich S, Frieling H, Walter M. Epigenetic Effects of Intravenous Diacetylmorphine on the Methylation of POMC and NR3C1. Neuropsychobiology. 2017;75(4):193-199. DOI: 10.1159/000486973. Epub 2018 Mar 6. PMID: 29510398.'.

Line 65, please change the font of the word 'in-utero' in italics font.

Line 101-103, Please detail the kind of method you have used to measures glucose intolerance.

Line 283-265, Please highlight DNA methylation guide only the changing in weight and growth patterns of offspring, and not the body composition, citing the supplementary table S1.

Line 301, Please in the limitations of your work insert that for measuring body composition are not used strong measures method like Bod Pod or DEXA or NMR.

Author Response

Many thanks for the reviewer for their positive feedback about this manuscript. Please see our responses below in red font:

The suggestion are the following:

Line 18, please change 'control' in 'controlled'.

Response to reviewer: Thank you for this comment, we have made this change in the manuscript (see abstract, line 18).

Line 31-32, Please insert drugs in the environmental factors that promote epigenetic changes in DNA, for your knowledge check this article: ' Groh A, Rhein M, Buchholz V, Burkert A, Huber CG, Lang UE, Borgwardt SJ, Heberlein A, Muschler MAN, Hillemacher T, Bleich S, Frieling H, Walter M. Epigenetic Effects of Intravenous Diacetylmorphine on the Methylation of POMC and NR3C1. Neuropsychobiology. 2017;75(4):193-199. DOI: 10.1159/000486973. Epub 2018 Mar 6. PMID: 29510398.'.

Response to reviewer: Many thanks for this comment and for bringing this paper to our attention. One of the reviews cited also covers the impact of drugs on epigenetic changes so we added drugs to the list of examples of environmental factors that promote epigenetic changes (See lines 31-32)

Line 65, please change the font of the word 'in-utero' in italics font.

Response to reviewer: Thank you for this comment, we have made this change in the manuscript (see introduction, line 66).

Line 101-103, Please detail the kind of method you have used to measures glucose intolerance.

Response to reviewer: Thank you for this comment. Glucose intolerance was measured with a glucose challenge test. We have added this information into the manuscript (see lines 104-106).

Line 283-265, Please highlight DNA methylation guide only the changing in weight and growth patterns of offspring, and not the body composition, citing the supplementary table S1.

Response to reviewer: Thank you for this comment. We have clarifying that the results refer to change in weight and growth patterns rather than body composition measures. (See lines 300-303; Our results suggest that change in weight and growth patterns, rather than body composition at one timepoint, in offspring may be guided by DNA methylation, particularly in early life, with our findings of weight centiles and BMI at 6 months of age being associated with some variation in methylation status in a subset of the sample group.)

Line 301, Please in the limitations of your work insert that for measuring body composition are not used strong measures method like Bod Pod or DEXA or NMR.

Response to reviewer: Thank you for this comment. We have added to our limitations that more sensitive measures of body composition such as air displacement plethysmography or dual-energy X-ray absorptiometry may be more suitable methods to detect subtle differences in body composition (see lines 327-329)